# *Saccharomyces cerevisiae* and *Clostridium butyricum* Could Improve B-Vitamin Production in the Rumen and Growth Performance of Heat-Stressed Goats

**DOI:** 10.3390/metabo12080766

**Published:** 2022-08-19

**Authors:** Liyuan Cai, Rudy Hartanto, Qingbiao Xu, Ji Zhang, Desheng Qi

**Affiliations:** 1College of Animal Sciences and Technology, Huazhong Agricultural University, Wuhan 430070, China; 2Department of Animal Science, Faculty of Animal and Agricultural Sciences, Diponegoro University, Semarang 50275, Indonesia

**Keywords:** goats, heat stress, *Clostridium butyricum*, *Saccharomyces cerevisiae*, B vitamins, growth performance

## Abstract

Heat stress can adversely affect the rumen environment and the growth performance of goats. The present study aimed to investigate the effects of *Saccharomyces cerevisiae* (SC), *Clostridium butyricum* (CB), and their mixture on B-vitamin production in the rumen and the growth performance of heat-stressed goats. Firstly, twelve Macheng × Boer crossed goats (24.21 ± 2.05 kg, control) were modeled to become heat-stressed goats (HS1). Then, the B-vitamin concentrations in the rumen and the parameters of growth performance were measured in goats. The results showed that heat stress could cause significantly decreased vitamin B1, B2, B6, B12, and niacin concentrations (*p* < 0.05). It also could cause a significantly reduced dry matter (DM) intake (DMI) and average daily gain (ADG) (*p* < 0.05). However, the digestibilities of DM, neutral detergent fiber (NDF), and acid detergent fiber (ADF) were significantly increased (*p* < 0.05) in HS1 compared to controls. Then, these twelve heat-stressed goats were divided equally into four groups: control group (HS2, no probiotic supplemented), SC group (0.30% SC supplemented to the basal diet), CB group (0.05% CB supplemented to the basal diet), and mix group (0.30% SC and 0.05% CB supplemented to the basal diet). They were used in a 4 × 4 Latin square experimental design. The results showed that the concentrations of vitamins B1, B2, and niacin in the rumen and the DMI, ADG, and the digestibility of DM, NDF, and ADF were significantly increased (*p* < 0.05) with SC, CB, and their mixture supplementation (*p* < 0.05). These results suggest that dietary supplementation with SC and CB could improve B-vitamin production in the rumen and the growth performance of heat-stressed goats.

## 1. Introduction

Ruminants are heat-intolerant livestock because a mass of heat is produced by rumen fermentation [1]. With global warming and intensive farming development, heat stress has become an urgent problem that needs to be solved in goat production in regions with high air temperatures and humidity in summer [2]. Heat stress could bring a number of adverse effects on rumen fermentation. Previous studies reported that when ruminants experience heat stress, the rumen pH and the concentration of rumen volatile fatty acids decrease significantly, and the ammonia N concentration is decreased [1,2,3,4,5]. These negative influences on rumen fermentation will eventually produce reductions in the overall performance of livestock.

In ruminants, the B vitamins are synthesized by certain ruminal microbial taxa and are involved in various processes related to nutritional metabolism, including the synthesis of fatty acids, the catabolism of branched-chain amino acids (BCAA), and gluconeogenesis [6,7]. A previous study reported that as the environment changes and the production performance of ruminants improves, the B-vitamin production is unlikely to meet the needs of the host itself [8]. However, few studies have reported the effects of heat stress on B-vitamin production in the rumen. When heat stress occurs in ruminants, their nutritional status is altered. Therefore, it is necessary to determine whether the B-vitamin production in the rumen is influenced by heat stress.

Probiotics are live microbial additives widely used worldwide to improve feed digestion, alimentation, growth performance, and health status [9,10]. A previous study reported that supplementation with probiotics in the diet can effectively relieve the adverse effects of heat stress in ruminants [11]. Our previous studies suggested that *Saccharomyces cerevisiae* (SC), *Clostridium butyricum* (CB), and their mixture could improve the rumen fermentation of heat-stressed goats. However, the effects of SC and CB on rumen B vitamins was not determined. This study aimed to investigate the effects of SC, CB, and their mixture on B-vitamin production in the rumen and the growth performance of heat-stressed goats. This study will provide a scientific reference for alleviating the adverse effects of heat stress on goat production.

## 2. Materials and Methods

### 2.1. Goats, Diet, and Management

Twelve female Macheng Black × Boer crossbred goats, aged 6.0 ± 1.0 months with a body weight of 24.21 ± 2.05 kg, were raised in individual pens (1.0 × 1.50 m) of a house equipped with slatted floors and manure scraper systems [12]. Goats were fed with a 1.31 kg/day basal diet. The ingredients and nutritional levels of the diet are given in Table 1. Our previous study showed that the water intakes were 2.10 L and 3.50 L during the non- and heat stress periods, respectively. Therefore, in this study, during the non-heat stress period, goats had free access to water and were gavaged with water to ensure that daily water intake reached 3.50 L, so as to achieve the same amount of drinking water as in the period of heat stress. No antibiotics or probiotics were given to these goats previously.

### 2.2. Modeling of Heat-Stressed Goats

As described by Cai et al. [2], the modeling processes of heat-stressed goats were divided into two periods (control and heat stress, HS1), and each period lasted 14 days. In the control period, the temperature was 25.0 ± 1.2 °C and the relative humidity was 62.1 ± 1.1% in the goat house. In the HS1 period, the temperature was 33.5 ± 2.5 °C and the relative humidity was 74.5 ± 2.4%. An air conditioner, an air heater, and a water sprinkler were applied to control the temperature and humidity of the goat house. The temperature–humidity index (THI) was used as an indicator for the evaluation of heat stress in goats, and it was used as an indicator of heat stress in goats in LPHSI (1990) [13] and was calculated as:THI = db℉ − (0.55 − 0.55 RH) (db℉ − 58)
where db℉ is the dry bulb temperature (℉), and RH is the relative humidity (%). In the HS1 period below, the THI was 87.5. Under this THI condition, these goats were considered as heat-stressed goats [14]. 

The blood and rumen fluid samples were collected on the last day of each period. In the morning, after 24 h fasting, blood samples were taken from the jugular veins of all the goats. After 4 h of feeding in the morning, the rumen fluids were collected from all the goats using a soft plastic stomach tube with a Jinteng GM-0.33A vacuum pump (Tianjin, China). Then, four layers of gauze were used to filter the rumen fluids to obtain rumen fluid without feed residue impurities. In the morning on days 11 to 13 of each period, five g of chromium (III) oxide (as an exogenous indicator) was added to the diet for the determination of the feed digestibility. Fecal samples were collected before morning feeding from the rectum of all the goats on days 12 to 14, and the fecal samples in the same period were pooled. Samples were stored at −20 °C for further analysis.

### 2.3. Probiotic Feeding Trials

The twelve heat-stressed goats were randomly divided into four groups and used in a 4 × 4 Latin square experimental design. The groups were the control group (HS2: fed the basal diet), *Saccharomyces cerevisiae*-supplemented group (SC: 0.30% *Saccharomyces cerevisiae* added to the basal diet), *Clostridium butyricum*-supplemented group (CB: 0.05% *Clostridium butyricum* added to the basal diet), and the mixture-supplemented group (mix: 0.30% *Saccharomyces cerevisiae* and 0.05% *Clostridium butyricum* added to the basal). The *Saccharomyces cerevisiae* was obtained from Angel Yeast Co., Ltd. (Yichang, China) and had a content of 2.0 × 10^10^ CFU/g live cells. The *Clostridium butyricum* was obtained from Huijia Biotechnology Co., Ltd. (Huzhou, China) and had a content of 1.0 × 10^8^ CFU/g live cells. There were four cycles in the feeding trial, and each cycle lasted for 20 days. In the morning on days 17 to 19 within each cycle, five g of chromium (III) oxide was added to the diet for the determination of feed digestibility. Between experimental cycles, all goats were fed a basal diet for 20 days to eliminate the influence of the previous treatment. On days 18 to 20 within each experimental cycle, fecal samples were collected from each goat. The rumen fluids were collected on day 20, 4 h after the morning feed. The methods of collection, pretreatment, and storage of rumen fluids were consistent with those stated in Section 2.2.

### 2.4. Measurement

The physiological parameters, including skin temperature, rectal temperature, respiratory rate, and pulse, were measured as described by Cai et al. [2]. The gene expression levels of the heat shock protein 70 (HSP70) and cortisol concentration were measured in blood [15,16]. Blood lymphocytes were isolated from whole blood by a peripheral blood lymphocyte isolation solution kit from Solarbio Science & Technology (Beijing, China). Total RNA of lymphocytes was extracted by TRIzol^®^ (Life Technologies, Carlsbad, CA, USA). A Revert Aid First Strand cDNA Synthesis kit from Thermo Fisher Scientific (Waltham, MA, USA) was used for reverse transcription. Primers were described in Cai et al. [17] and were synthesized by Sangon Biotech Co., Ltd. (Shanghai, China). The primer sequences are given in Table 2. A SYBR RT-PCR Kit from Bio-Rad (Hercules, CA, USA) in conjunction with an ABI QuanStudio TM6 flex real-time fluorescent quantitative PCR system (Life Technologies, Carlsbad, CA, USA) were used for the PCR procedure. Each sample was analyzed in triplicate. The relative expression levels were quantified using the 2^−∆∆Ct^ method [18]. Blood in collection tubes was centrifuged at 3000 rpm for 10 min to obtain serum. Serum cortisol was measured by a cortisol assay kit from Nanjing Jiancheng Bioengineering Institute (Nanjing, China), following the manufacturer’s instructions. 

Vitamins B1, B2, and B6 were measured according to Ciulu et al. [19]. Vitamin B12 and niacin were measured according to China National Standard GB/T 17819-1999 and China National Standard GB/T 5009 197-2003, respectively. In brief, vitamins B1, B2, B6, B12, and niacin in rumen fluids were measured by a 2100 liquid chromatograph (Shimadzu, Japan). Separation was performed on an Agilent Eclipse XDB-C18 column, 250 mm × 4.6 mm, 5 μm particle size (Agilent Technologies, Santa Clara, CA, USA). Each sample was prepared and injected in triplicate. The injection volume was 20 μL. For vitamins B1, B2, and B6 detection, mobile phase A was 0.025% trifluoroacetic acid, and mobile phase B was acetonitrile; A:B= 9:1 (*v*/*v*). The flow rate of the mobile phase was 1.0 mL/min, and the column temperature was 25 °C, λ = 280 nm. For vitamin B12 detection, mobile phase A was 2.0% phosphoric acid, and mobile phase B was acetonitrile; A: B = 9:1 (*v*/*v*). The flow rate of the mobile phase was 1.0 mL/min, and the column temperature was 30 °C, λ = 361 nm. For niacin acid detection, mobile phase A was the mixture of 10 mL glacial acetic acid, 1.3 mL triethylamine, 20 mL 0.005 mol/L sodium heptanesulfonate, and deionized water diluted to 1 L (pH = 3.2). The flow rate of the mobile phase was 1.0 mL/min, and the column temperature was 25 °C, λ = 275 nm. Before use, all the mobile phases were filtered with a 0.22 μm membrane and ultrasonic degassing for 20 min.

The determination of dry matter (DM) in feedstuff and fecal matter referred to method #930.15 in AOAC (2005) [20]. As described by Goering and Van Soest (1970) [21], neutral detergent fiber (NDF) and acid detergent fiber (ADF) were measured in feedstuff and fecal matter. Daily matter intake (DMI) were recorded every day to calculate the average values. The body weights were recorded at the beginning and end of each period or experimental cycle to allow average daily gain (ADG) calculations.

### 2.5. Statistical Analysis

The data of this study were analyzed by R studio (v4.0.5, GitHub Inc., San Francisco, CA, USA). Data of control and HS1 periods were analyzed using a two-tailed Student’s *t*-test for each significant factor or interaction. Data of HS2, SC, CB, and mix were analyzed using two-way analysis of variance (ANOVA) tests followed by post hoc Dunn test for each significant factor or interaction. *p* values of less than 0.05 were considered statistically significant.

## 3. Results

### 3.1. Successfully Modeling Heat-Stressed Goats

Under the condition of THI of 87.5, to further confirm the occurrence of heat stress, the expression of HSP 70 genes in the blood lymphocytes and the physiological parameters of goats was determined. The expression of HSPA 1 was significantly increased (*p* < 0.05; Figure 1A), while there were no differences in the expression of HSPA 6 and HSPA 8 in the blood lymphocytes between control and HS1 animals (*p* > 0.05; Figure 1A). Moreover, serum cortisol concentrations were significantly increased in HS1 compared to control animals (*p* < 0.01; Figure 1B). The skin temperatures, respiratory rates, and pulses were significantly increased (*p* < 0.05). However, there were no significant differences in the rectal temperatures in HS1 compared to control animals. The physiological parameters of the goats are given in Table 3. 

### 3.2. Heat Stress Caused a Significant Decrease in Rumen B-Vitamin Concentration

The concentrations of vitamins B1, B2, B6, B12, and niacin were significantly decreased in the rumen fluid of HS2 compared with control animals (*p* < 0.05; Figure 2). 

### 3.3. Heat Stress Caused a Significant Decrease in Growth Performance

The dry matter intake (DMI) and average daily gain (ADG) were significantly decreased (*p* < 0.05), while the digestibilities of DM, NDF, and ADF were significantly increased (*p* < 0.05) in HS1 compared to control goats. The growth performance parameters of the goats in the non- and heat stress periods (control vs. HS1) are given in Table 4.

### 3.4. SC and CB Improved B-Vitamin Production in the Rumen of Heat-Stressed Goats

Vitamins B1, B2, and niacin concentrations were significantly increased in the rumen fluid of SC, CB, and mix groups compared with that of HS2 (*p* < 0.05; Figure 3). 

### 3.5. SC and CB Improved Growth Performance of Heat-Stressed Goats

SC, CB, and mix goats exhibited significantly increased (*p* < 0.05) ADG, and the digestibilities of DM, NDF, and ADF were significantly increased (*p* < 0.05) compared with those in HS2. The growth performance parameters of the heat-stressed goats supplemented with CB and SC are given in Table 5.

## 4. Discussion

Excessive heat was produced from rumen fermentation, which led to the ruminants having a low tolerance to heat [1]. It is generally believed that goats are not prone to heat stress. However, our previous research suggested that goats live in a thermal environment where the THI is higher than 82 for 14 days. Heat stress occurs and brings a series of adverse effects on rumen fermentation and the growth performance of goats [2].

In ruminants, the B vitamins function as enzyme cofactors or precursors for cofactors [22]. Traditional ruminant nutrition theory holds that the rumen microbiota can produce B vitamins that satisfy the needs of the ruminant host. However, more and more studies show that, in some physiological or rearing conditions, ruminants have a B-vitamin deficiency [6,23]. It is reported that vitamin B1 deficiency occurs when ruminants ingest a high concentrate diet and cause ruminal acidosis (SARA). Then, supplementing the diet with thiamine can increase the rumen pH, and the relative abundance of cellulose-decomposing bacteria such as rumen *Bacteroides*, *Ruminococcus*, and *Succinivibrio* increases to improve the digestibilities of fiber in the rumen [24,25]. Vitamin B12 deficiency occurred during lactation in dairy cows, and the intake of foodstuffs increased after vitamin B12 injection [26]. Supplementing with niacin at 19 g/day in the diet for heat-stressed cows, the number of somatic cells in milk was significantly decreased [27]. Most of the previous studies have focused on the improvement of ruminant production performance and health status by supplementing exogenous B vitamins and have not evaluated the impact on rumen B-vitamin production in special physiological periods or sub-health status. The present study compares the production of B vitamins in the rumen of goats during non- and heat stress periods. We found that the concentrations of vitamins B1, B2, B6, and niacin were significantly decreased in rumen fluids when the goats were exposed to heat stress. The decrease in B vitamins may be due to the change in rumen fermentation during heat stress [2]. Therefore, breeders should pay attention to B-vitamin deficiencies in goats caused by heat stress, and we should choose suitable methods for B-vitamin supplementation.

Probiotics have been widely used for many years to enhance feed digestion and supplement the diet, being effective in relieving the adverse effects of heat stress in livestock production [11]. Few studies have reported the effects of probiotics on rumen B-vitamin production. In this study, supplementation with *Saccharomyces cerevisiae*, *Clostridium butyricum*, and their mixture could enhance the concentrations of vitamin B1, B2, and niacin in the rumen of heat-stressed goats. Such an effect may be because yeast cells contain glucose, furan mannose, and chitin, which can be used as fermentation substrates for rumen microbiota to promote the production of B vitamins [28,29]. Moreover, the yeast is rich in B vitamins, which also directly provides B vitamins for rumen microbiota and ruminants [30]. *Clostridium butyricum* can produce B vitamins in the digestive tracts of animals, which also provides B vitamins for the rumen microbiota and goats [31,32]. In this study, these two probiotics promoted the production of vitamins B1, B2, and niacin in the rumen of heat-stressed goats, which may be because the two probiotics corrected the imbalance of the rumen microbiota community and partially restored the ability of the microbiota to produce B vitamins. In addition, they supplement B vitamins by themselves by containing and producing them.

The results of this study also indicate that the supplementation of probiotics cannot completely correct the B-vitamin deficiency caused by heat stress. Therefore, it is necessary to consider the use of more effective methods to correct the B-vitamin deficiency during the heat stress period.

A previous study has reported that during the heat stress period, ADG decreases significantly in ruminants [2,33,34,35]. The results of ADG in this study are consistent with the findings of previous studies. The decreased ADG may be caused by the adverse effects of heat stress on nutrition metabolism and enzymatic reactions [1]. B vitamins play essential roles in several metabolic processes [7], and the decrease in B-vitamin production in the rumen may be a key factor affecting ADG. However, this speculation still needs to be validated in future studies.

## 5. Conclusions

Heat stress brings adverse effects on B-vitamin production in the rumen, in which the concentrations of vitamins B1, B2, B6, B12, and niacin are significantly decreased. Moreover, heat stress also negatively influences growth performance, in which the DMI and ADG are significantly decreased. *Saccharomyces cerevisiae*, *Clostridium butyricum*, and their mixture were beneficial for heat-stressed goats to enhance the B-vitamin production in the rumen and improve the growth performance of the animals. Therefore, supplementation with *Saccharomyces cerevisiae* and *Clostridium butyricum* can effectively relieve the adverse effects of B-vitamin production in the rumen of goats and their growth performance.

## Figures and Tables

**Figure 1 metabolites-12-00766-f001:**
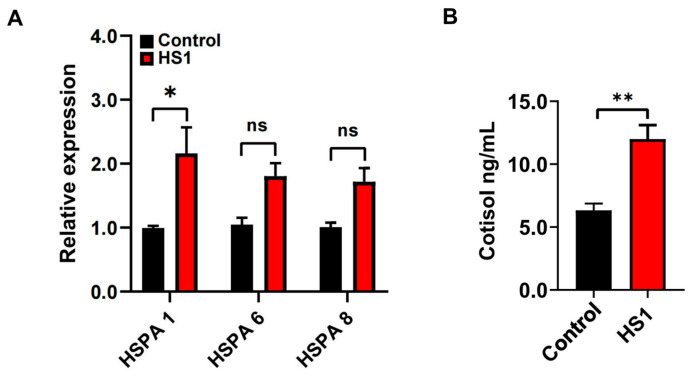
The expression of Hsp 70 genes and the concentrations of serum cortisol in goats. (**A**) The expression of HSPA 1, HSPA 6, and HSPA 8 genes in the blood lymphocytes of control and HS1. (**B**) The concentrations of serum cortisol of control and HS1. Data were analyzed using a two-tailed Student’s *t*-test and were considered statistically significant at * *p* < 0.05 and ** *p* < 0.01 between the control and HS1 groups. Data are expressed as the mean ± SEM. “ns” means not significant.

**Figure 2 metabolites-12-00766-f002:**
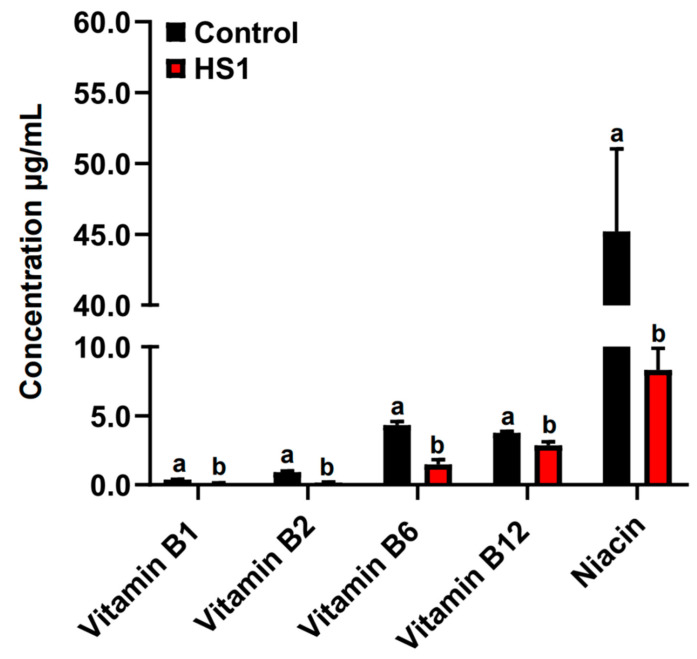
The concentrations of vitamins B1, B2, B6, B12, and niacin in the rumen fluid of control and HS2 animals. The different letter indicated significantly different (*p* < 0.05) of the same B-vitamin.

**Figure 3 metabolites-12-00766-f003:**
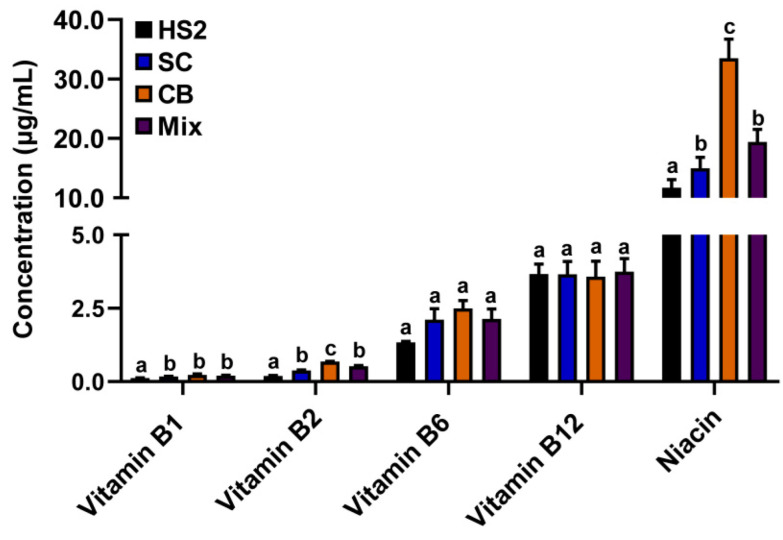
Vitamins B1, B2, B6, and niacin concentrations in rumen fluid of goats with CB, SC, and their combination supplementation. The different lower-letter indicated significantly different (*p* < 0.05) and the same lower-letter indicated no significantly different (*p* > 0.05) of the same B-vitamin.

**Table 1 metabolites-12-00766-t001:** The ingredients and nutritional composition (g/kg) of the diet.

Ingredient	Content	Nutrition Level	Content
Alfalfa	562	Dry matter	951
Ground corn	264	Organic matter	854
Soybean meal	84	Crude protein	173
Wheat barn	73	Neutral detergent fiber (NDF)	434
Ca_2_HPO_4_	7	Acid detergent fiber (ADF)	257
Premix *	10	Calcium	5.9
		Phosphorus	3.2

* Premix contained per kg: 20.70 g Mg, 0.50 g Fe, 1 g Mn, 2 g Zn, 43 mg Se, 47 mg I, 54 mg, Co, 90,000 IU vitamin A, 17,000 IU vitamin D, 1750 IU vitamin E.

**Table 2 metabolites-12-00766-t002:** Details of the primer sequences.

Gene	Primer Sequence	Product Length	Annealing Temperature	GenBank Accession No.
β-actin	F:TCTGGCACCACACCTTCTAC R: TCTTCTCACGGTTGGGCCTTG	102	60	XM_018039831.1
HSPA 1	F:CGACCAGGGAAACCGGCAC R:CGGGTCGCCGAACTTGC	151	60	NM_005677146.3
HSPA 6	F:TCTGCCGCAACAGGATAAA R:CGCCCACGCACGAGTAC	239	60	NM_001314233.1
HSPA 8	F:ACCTCTATTACCCGTGCCC R:CTCTTATTCAGTTCCTTCCCATT	203	60	XM_018039831.1

**Table 3 metabolites-12-00766-t003:** Influence of heat stress on the physiological parameters of goats.

Parameters	Periods		SEM
	Control	HS1	
Skin temperature (°C)	33.8 ^a^	36.8 ^b^	0.23
Rectal temperature (°C)	39.1 ^a^	39.6 ^a^	0.17
Respiratory rate (breaths/min)	25.3 ^a^	33.4 ^b^	1.12
Pulse (beats/min)	74.6 ^a^	84.1 ^b^	3.22

^a,b^ Means within a row with different superscripts differ significantly (*p* < 0.05).

**Table 4 metabolites-12-00766-t004:** The growth performance parameters of the goats in non- and heat stress periods.

Parameters	Periods		SEM
	Control	HS1	
DMI (kg)	1.13 ^a^	0.84 ^b^	0.04
ADG (kg)	0.13 ^a^	0.08 ^b^	0.02
DM (%)	51.28 ^a^	54.54 ^b^	3.95
NDF (%)	40.14 ^a^	47.04 ^a^	3.13
ADF (%)	39.87 ^a^	44.28 ^b^	3.22

^a,b^ Means within a row with different superscripts differ significantly (*p* < 0.05).

**Table 5 metabolites-12-00766-t005:** The growth performance parameters of the heat-stressed goats supplemented with *Clostridium butyricum* and *Saccharomyces cerevisiae*.

Parameters	Treatment				SEM
	HS2	CB	SC	Mix	
DMI (kg)	0.81 ^a^	0.85 ^a^	0.82 ^a^	0.84 ^a^	0.03
ADG (kg)	0.08 ^a^	0.17 ^b^	0.12 ^c^	0.12 ^c^	0.01
DM (%)	50.52 ^a^	66.02 ^b^	57.23 ^c^	65.21 ^b^	4.23
NDF (%)	38.34 ^a^	54.24 ^b^	46.24 ^c^	52.44 ^a^	3.11
ADF (%)	37.64 ^a^	50.14 ^b^	44.32 ^c^	49.02 ^b^	3.01

^a–c^ Means within a row with different superscripts differ significantly (*p* < 0.05).

## Data Availability

No new data were created or analyzed in this study. Data sharing is not applicable to this article.

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
