# Peer review of "Saccharomyces cerevisiae and Clostridium butyricum Could Improve B-Vitamin Production in the Rumen and Growth Performance of Heat-Stressed Goats"

_metabolites, 2022, doi:10.3390/metabo12080766_

Round 1

Reviewer 1 Report

A very interesting manuscript with practical implications, but:

Chapter Material and Methods needs to be improved, it is very chaotic. No description of the CG group, only in Abstract and Results

and:

Line 28 - please delete: "were significantly increased"

Line 46 - please explain the abbreviation BCAA

Line 120-122 - please delete "Before the morning and afternoon feedings, fecal samples were collected from each goat on days 18 to 20 within each experimental cycle. The rumen fluids  were collected on day 20, 4 h after the morning feed." 

Line 141-142 - PBLs from serum or whole blood? there is no information about whole blood collection from goats.  

Line 239-240 and 44-45 the same information but different citations ?

Author Response

Dear reviewer,

Thank you for your email from 8 August 2022 on our manuscript (Manuscript ID: metabolites-1855063), entitled “Dietary Supplementation with Clostridium butyricum and Saccharomyces cerevisiae Improved B-vitamins Production in the Rumen and Growth Performance of Heat-Stressed Goats (The original title has been changed to “Saccharomyces cerevisiae and Clostridium butyricum could improved B-vitamins Production in the Rumen and Growth Performance of Heat-stressed Goats”) by Liyuan Cai et al. We are truly grateful for the reviewers’ comments and suggestions on improving the manuscript, which helped us greatly improve the quality of our paper. Based on their suggestions, we have made careful modifications to the original manuscript in both the content of the manuscript and the language, and efforts were also made to revise many improper expressions. The manuscript is now more readable for readers, and we hope that the revised manuscript will meet the standards for publication. Here submitted is a revised manuscript.

It would be greatly appreciated if Metabolites could accept our paper for publication. We look forward to hearing from you.

Sincerely yours,

Liyuan Cai

Reviewer 2 Report

This manuscript tried to investigate the effect of Clostridium butyricum (CB) and Saccharomyces cerevisiae (SC) on B-vitamin production and growth performance of heat-stressed goat. Authors clearly showed that CB and SC improve heat stress induced reduction of B-vitamins production and growth. However, I have some concern as follows in this manuscript.

1. In Abstract and Conclusion

Authors concluded that SC, CB, and their combination could alleviate the decrease of B-vitamins production. I understand thar SC and CB have those effects. However, I could not understand combination effects of those in this study. For example, authors did not show any additive effect on B-vitamins production and growth performance in combination group compared with those of CB or SC groups. Authors should change the conclusion In this point.

2. In Introduction page1, line 42.

Please mention how the ammonia-N concentrations is influenced. Increased or decreased?

3. In Methods and Results sections

Please mention detail information of CG and HS1 group. In figure 1, authors compared data from CG and HS1. I could not understand sample size and timing of sample correction (How were the samples taken?)

4. In Methods section

Please mention detail information of DM(%), NDF(%), and ADF(%). I could not easily access the reference [20]. And I also want to know what do DM(%), NDF(%), and ADF(%) mean?

5. In Results section

Please show raw data of body weight changes of this study. These can help to reproduce this study.

Author Response

(The authors gave the same response as above.)

Round 2

Reviewer 1 Report

Point 1. Line 44-45 : In ruminants, the B-vitamins are synthesized by the microbiota and function as enzyme cofactors or precursors for cofactors (6).

Line 225- 226 : In ruminants, the B-vitamins are synthesized by the microbiota and function as enzyme cofactors or precursors for cofactors [22].

Same sentence, different citations!!!

Point 2. in the first draft of the manuscript sent for review:

Line 118-121 and line 135-137 described the same action, so I suggested deleting the first fragment

In the current version the whole Material and Methods section has been changed

Author Response

Dear reviewer,

Thanks for your careful review and pertinent suggestions for our manuscript. Entitled “Saccharomyces cerevisiae and Clostridium butyricum Could Improve B-vitamins Production in the Rumen and Growth Performance of Heat-stressed Goats by Liyuan Cai et al. We are truly grateful for your suggestions on improving the manuscript, which helped us greatly improve the quality of our paper. Based on their suggestion, we have made careful modifications to the manuscript. The manuscript is now more readable for readers.

Sincerely yours,

Liyuan Cai